# Pathophysiology of Calcium Mediated Ventricular Arrhythmias and Novel Therapeutic Options with Focus on Gene Therapy

**DOI:** 10.3390/ijms20215304

**Published:** 2019-10-24

**Authors:** Vera Paar, Peter Jirak, Robert Larbig, Naufal Shamilevich Zagidullin, Mathias C. Brandt, Michael Lichtenauer, Uta C. Hoppe, Lukas J. Motloch

**Affiliations:** 1Department of Internal Medicine II, Paracelsus Medical University, 5020 Salzburg, Austria; v.paar@salk.at (V.P.); p.jirak@salk.at (P.J.); m.brandt@salk.at (M.C.B.); m.lichtenauer@salk.at (M.L.); u.hoppe@salk.at (U.C.H.); 2Division of Cardiology, Hospital Maria Hilf Moenchengladbach, 41063 Moenchengladbach, Germany; robert.larbig@mariahilf.de; 3Department of Internal Diseases, Bashkir State Medical University, 450000 Ufa, Russia; nau36@yahoo.com

**Keywords:** ventricular arrhythmias, calcium, ion channels, heart failure, genetic mutations, L-type calcium channel, ryanodine receptor, sarcoplasmic Ca^2+^-ATPase, mitochondria, gene therapy

## Abstract

Cardiac arrhythmias constitute a major health problem with a huge impact on mortality rates and health care costs. Despite ongoing research efforts, the understanding of the molecular mechanisms and processes responsible for arrhythmogenesis remains incomplete. Given the crucial role of Ca^2+^-handling in action potential generation and cardiac contraction, Ca^2+^ channels and Ca^2+^ handling proteins represent promising targets for suppression of ventricular arrhythmias. Accordingly, we report the different roles of Ca^2+^-handling in the development of congenital as well as acquired ventricular arrhythmia syndromes. We highlight the therapeutic potential of gene therapy as a novel and innovative approach for future arrhythmia therapy. Furthermore, we discuss various promising cellular and mitochondrial targets for therapeutic gene transfer currently under investigation.

Cardiac arrhythmias constitute a major public health concern worldwide [1]. Several severe types of arrhythmias are responsible for syncope and sudden cardiac death (SCD). Ventricular fibrillation and tachycardia are either based on genetic mutations concerning ion channel expression and function or on arrhythmogenic substrates that may be caused by heart failure (HF), infarction or drugs that modify ion channel behavior [2,3]. The incomplete understanding of arrhythmia formation and their underlying mechanisms in the failing heart has hindered the development of safe and effective pharmacological treatments [4]. In fact, some pharmacological agents even constitute a pro-arrhythmic tendency, which may lead to arrhythmia-related deaths. For instance, the Cardiac Arrhythmia Suppression Trial (CAST), which was terminated prematurely after a substantial increase in arrhythmic deaths of patients, treated with either encainide or flecainide compared to placebo [5]. These pro-arrhythmic effects may be based upon the heterogeneity of cardiac ion channel expression and function within the different regions and layers of the heart [6,7], the unspecific nature of multiple pharmacological agents targeting ion channels, the complexity of ion channel behavior and their cross-talk, or the dynamic cellular and environmental remodeling properties caused by cardiac disease progression [8].

Therefore, investigations on ionic currents and channels, and their influence on action potential (AP) waveforms are of major interest for understanding arrhythmogenesis. Calcium (Ca^2+^) homeostasis plays a crucial role in AP generation and correct cardiac contraction. Several studies have shown that cellular calcium handling is altered in the pathophysiology of heart diseases, such as HF and cardiomyopathies [9]. Additionally, Ca^2+^ handling may also be altered by inherited mutations that directly affect Ca^2+^ channels or Ca^2+^ handling proteins, such as ryanodine receptor type 2 (RyR2) [10], calsequestrin (CASQ) [11] and calmodulin (CaM) [12]. Furthermore, the role of myocardial mitochondria in Ca^2+^ handling gets more into focus of current studies [13,14,15]. In this review we will focus on ventricular arrhythmias caused by defective Ca^2+^ currents or channels and novel therapeutic options.

## 1. The Role of Calcium in Action Potential Generation

Cardiac contraction and relaxation are mediated by a precise and coordinated linkage of electrical activation (excitation) and intracellular Ca^2+^ homeostasis, resulting in a so-called excitation-contraction coupling. Each ventricular AP starts with the influx of sodium (Na^+^) through voltage-gated Na^+^ channels that depolarizes the cell. When a certain threshold is reached voltage-sensitive L-type Ca^2+^ channels (LTCC) are activated that allow the influx of Ca^2+^ into the cytosol, triggering the much larger Ca^2+^ release from the sarcoplasmic reticulum (SR), the main intracellular Ca^2+^ storage organelle [16,17]. SR Ca^2+^ release is mediated by RyR2, which is a massive structure comprising the largest known ion channel-bearing macromolecular complex [18,19]. This process is called Ca^2+^-induced Ca^2+^-release (CICR) and is the fundamental link between electrical and mechanical activation in the heart [20]. The cytosolic Ca^2+^ then binds to troponin C and initiates myocardial contraction, forming the cardiac systole [16].

During diastole, myofilaments relax, either due to calcium re-uptake into the SR by the SR Ca^2+^-ATPase type-2a (SERCA2a) that pumps Ca^2+^ back into the SR stores, or Ca^2+^ is extruded and released into the extracellular space through Na^+^/Ca^2+^ exchanger (NCX) that exchanges approximately three Na^+^ ions entering the cell in exchange for one Ca^2+^ ion leaving [21]. Thus, the NCX removes Ca^2+^ by generating a net inward depolarizing current, called the transient inward current (I_ti_). Approximately 63% of cytosolic Ca^2+^ are taken up by SERCA2a and around 37% Ca^2+^ are extruded by NCX in humans [22,23].

The depolarization and release of Ca^2+^ into the cytosol and the rapid re-uptake or extrusion results in a Ca^2+^ wave, which is known as the Ca^2+^ transient. The amount of Ca^2+^ released from the SR directly correlates with the Ca^2+^ transient amplitude and is responsible for the strength of systolic contraction [16]. Consequently, any perturbations in intracellular Ca^2+^ handling could result in changes of electrical stability and cardiac contractility, which may lead to malignant ventricular arrhythmias.

### 1.1. The Role of β-Adrenergic Receptor Stimulation

Additionally, the activation of the adrenergic nervous system through β-adrenergic receptor (β-AR) stimulation has a great impact on Ca^2+^ handling. β-AR activates a crucial cascade of events, resulting in the phosphorylation of Ca^2+^-mediating proteins. In principal, two effects follow β-AR signaling: (1) an increased activity of LTCC and (2) an enhanced Ca^2+^ concentration in the SR due to stimulation of SERCA2a [24]. Both channels are activated through phosphorylation activities of the two enzymes protein kinase A (PKA) [25] or Ca^2+^/calmodulin-dependent protein kinase type II (CaMKII) [26]. Hence, the emerging phosphorylation of LTCC increases the amplitude of the current. Additionally, phosphorylation of phospholamban (PLN) hampers its inhibiting properties on SERCA2a, resulting in an increased SR Ca^2+^ re-uptake. On the other hand, the activity of PKA and CaMKII phosphorylates RyR2 and causes an activation of the Ca^2+^ releasing channel. In physiological conditions, this phosphorylation property, due to β-AR stimulation, is an important mechanism to enable SR Ca^2+^ release. This leads to an improvement of myocardial contractility in response to environmental stressors [27,28]. The influence of β-AR stimulation in pathophysiological conditions is described in the following section.

### 1.2. Cardiac Mitochondrial Ca^2+^ (mCa^2+^) Handling

For correct AP progression and myocardial contraction, several adenosine triphosphate (ATP)-consuming proteins, or so-called ion-pumps, such as SERCA2a (as mentioned above), are necessary. Therefore, a continuous supply of energy, in form of ATP, is required and provided by oxidative phosphorylation in myocardial mitochondria [29]. Cardiac mitochondria occupy about 20–30% of the cardiomyocyte’s volume and are located near the SR [30]. In humans, it is estimated that approximately 6 kg (kg) of ATP are hydrolyzed in the heart per day [31]. Mitochondrial Ca^2+^ uptake from the cytosol directly correlates with the amount of energy being produced [32]. As the inner mitochondrial membrane is rather impermeable for almost all ions, Ca^2+^ uptake is thought to be mostly mediated by the mitochondrial Ca^2+^ uniporter (MCU) [13,33,34]. The channel is known to be blocked by nanomolar concentrations of ruthenium red (RuR) or by its more specific derivate, ruthenium 360 (Ru360). In addition, it has been suggested that adenine nucleotides tend to suppress MCU activity, with ATP being the most effective one. Consequently, if a huge amount of ATP is available, Ca^2+^ uptake and ATP synthesis in mitochondria is inhibited [33,35,36].

The MCU is a highly Ca^2+^-selective protein complex that consists of the pore-forming mitochondrial Ca^2+^ uniporter protein [36,37], the essential MCU regulator (EMRE), and the mitochondrial calcium uptake 1 and 2 (MICU1/2) [38,39,40,41,42]. Furthermore, besides the pore forming units, other mitochondrial proteins seem to modulate the MCU activity [13,43,44,45]. Indeed, the mitochondrial uncoupling proteins (UCPs) 2 and 3, which are located in the cardiac inner mitochondrial membrane, are suggested to influence MCU function in an ATP dependent manner [45,46,47,48,49,50]. This interaction results in UCP2-dependent modulatory effects on cardiac excitation contraction coupling via altered LTCC activity. LTCC amplitude was shown to be decreased in UCP knock out cardiomyocytes vs. control. The authors speculate this was possibly due to Ca^2+^-dependent inactivation of LTCCs during increased dyadic cleft Ca^2+^ in UCP2 knock out cardiomyocytes [45,51]. 

The mitochondrial NCX (mNCX or NCLX) constitutes the major mitochondrial Ca^2+^ efflux pathway in the heart [52]. Similar to the sarcolemmal form it transports stoichiometry 3 Na^+^:1 Ca^2+^ [53]. This channel is essential to prevent mitochondrial Ca^2+^ overload with consequent effects on ROS generation [54]. As it is the case for sarcolemmal NCX, also a reverse mode for NCLX has been described. These effects may play an important role in diseased hearts with known alterations in Na^+^ and Ca^2+^ concentrations, such as HF [34,55,56]. 

With respect to cytosolic Ca^2+^ handling, there is still considerable debate whether mitochondrial Ca^2+^-uptake might affect cytosolic Ca^2+^-homeostasis [34,45]. 

## 2. Pathophysiology of Calcium Mediated Ventricular Arrhythmia

Malignant ventricular tachyarrhythmias are caused by either (1) enhanced automaticity, (2) triggered activity or (3) reentry. The first two conditions refer to cellular phenomena, whereas the latter is a matter of the cardiac electrophysiological network [57,58,59].

Firstly, “enhanced automaticity” is the acceleration of the spontaneous firing rate of cardiomyocytes (CMs). If ventricular CMs, normally quiescent, automatically generate APs due to disease or genetic modifications, irregular activation patterns arise. These abnormal wave fronts then collide against or compete with normal waves originating from the sinoatrial node. Secondly, “triggered activity” refers to Ca^2+^-mediated premature APs that result from early or delayed afterdepolarization (EADs, or DADs). These premature cellular depolarizations may provoke ventricular contraction or short runs of ventricular tachycardia (VT). The third and most common manifestation is “reentry” where one or several wave fronts circulate around zones of refractory tissue or rotate as spiral waves [58,59,60,61,62].

Pathophysiological conditions resulting in Ca^2+^-mediated ventricular arrhythmia and VT may either be caused by heritable mutations in cardiac ion channels and channel-interacting proteins or develop through acquired diseases of the myocardium (Figure 1).

### 2.1. Spontaneous Ca^2+^ Release Events (SCRs)

In SCRs, Ca^2+^ release is not triggered by an AP, but evoked by channelopathies (pathophysiological remodeling and mutations in ion channels) or due to acquired conditions, such as chronic HF, injury from myocardial ischemia or ischemia reperfusion (I/R) [63]. The probability of SCR dependent Ca^2+^ waves is related to the balance between the SR Ca^2+^ concentration and the minimum level of Ca^2+^, which is able to induce Ca^2+^ release from the SR, the so-called calcium threshold [64]. RyR2 and its inhibitory protein calsequestrin 2 (CASQ2) play the crucial roles in maintaining this threshold.

Notably, in inherited genetic mutations, such as catecholaminergic polymorphic VT (CPVT), the fundamental step of arrhythmogenesis is SCR due to mutations in either RyR2 (CPVT-1) or CASQ2 (CPVT-2) [65,66]. Several studies have shown that both, mutations in RyR2 [67,68] or CASQ2 [69], cause a significant increase in RyR2 activity, facilitating the occurrence of SCRs.

However, β-AR activity seems to be crucial for the generation of SCRs. As mentioned above, β-AR is responsible for the activation of LTCCs and constitutes a trigger for SR Ca^2+^ re-uptake due to SERCA2a phosphorylation. However, in pathological conditions, the RyR2 channel can be chronically hyperphosphorylated. Consequently, channel opening probability increases, leading to diastolic Ca^2+^ release, the so-called Ca^2+^ leak, from the SR [70,71]. However, even if activation of RyR2 increases, arrhythmias are only observed during β-AR stimulation [72]. On the other hand, some studies have elucidated PKA- and CaMKII-related phosphorylation of RyR2 to be the essential arrhythmogenic step [73]. Of note, CaMKII increases LTCC activity, thus removes the inhibitory effect of PLN on SERCA2a, and activates RyR2 [74]. LTCC activity is also affected by cardiac remodeling processes. In their elegant trials, Brooksby et al. and Cerbai et al., revealed that Ca^2+^ channel expression is highly dependent on the stage of HF [75,76], as the density of LTCCs is elevated in mild to moderate hypertrophy and decreases in more severe stages of hypertrophy and HF [77,78]. Of note, CMs from failing hearts exhibit a slight alteration in responsiveness to β-AR stimulation, resulting in slowing of LTCC inactivation and AP prolongation [79,80,81].

Furthermore, mutations and genetic defects in the genes CACNA1C, CACNB2, and CACNA2D1 (encoding components of LTCC) may result in various arrhythmogenic syndromes like the Brugada syndrome [82,83], Brugada syndrome with short QT duration [83,84], short QT syndrome [83,84,85], early repolarization syndrome [82,86] and idiopathic ventricular fibrillation (VF). All these mutations result in a loss-of-function of LTCCs [87].

### 2.2. Action Potential Prolongation and Afterdepolarizations

In the diseased heart, due to pathophysiological alterations like HF or genetic mutations, cellular remodeling is often accompanied by changes in Ca^2+^ channel expression and function. When abnormal Ca^2+^ release occurs, intracellular Ca^2+^ levels increase, potentially leading to cytosolic Ca^2+^ overload. To maintain Ca^2+^ homeostasis counterbalancing mechanisms, such as the NCX channel, need to be enhanced [88]. This current in turn produces transient membrane depolarization at the plateau phase or after the completion of the AP, leading to EADs or DADs, respectively [89]. When a certain threshold is reached at the plateau phase of the AP, due to EAD amplitude, LTCC channels are again activated, leading to AP prolongation. The longer the AP duration, the higher the possibility for EADs to occur due to the recovery of LTCCs from inactivation [61]. The generated inward current leads to a secondary membrane depolarization that interrupts the repolarization phase of the previous beat. This can further prolong AP and may be the origin for more EADs. In case of DADs, the net inward current generated by SCRs or NCX activity may trigger Na^+^ channel opening, resulting in a premature beat. Furthermore, if a certain threshold is reached, the electrical wave could spread to downstream cells and trigger premature contraction [90]. Of note, EADs, which occur during the so-called vulnerable window of repolarization, have a great potential to initiate sustained arrhythmias by reentrant mechanisms [91]. However, these events are mostly triggered by the presence of a pathologic electrophysiological substrate, which enables the formation of a unidirectional conduction block. Most commonly fibrotic formations and action potential duration (APD) gradients, which are present in the failing heart, promote such reentrant mechanisms [90].

### 2.3. The Role of Mitochondria in Ca^2+^ Handling of Diseased Heart

As mentioned above, dysregulation of Ca^2+^ handling, leading to a reduced cytosolic Ca^2+^ transient during excitation on the one hand but increased baseline cytosolic Ca^2+^ level on the other hand is a hallmark of HF. Since mitochondria are located in close proximity of the SR, they are speculated to act as a cytosolic Ca^2+^ sink in cardiac pathologies. The MCU has a low affinity but a high capacity to transport Ca^2+^. Therefore, under physiological conditions a rapid uptake of cytosolic Ca^2+^ into the mitochondria at the plateau phase, and not under basal conditions of the AP, is accomplished. Consequently, with increased cytosolic Ca^2+^ level at baseline, MCU related Ca^2+^ uptake could play a crucial role in balancing Ca^2+^ homeostasis in the failing heart [92,93]. However, this mechanism might promote mitochondrial Ca^2+^ overload and contribute to mitochondrial dysfunction [94,95].

Furthermore, since mitochondrial Ca^2+^ handling regulates mitochondrial energy generation, further Ca^2+^ dependent alterations might be suggested in mitochondria from the diseased heart. Besides ATP, also reactive oxygen species (ROS) emerge through the reaction of electrons with oxygen, forming superoxide anions (O_2_^−^) [96]. In healthy cardiac tissue, redox balance is maintained by efficient antioxidant mechanisms that prevent excessive ROS accumulation. However, in diseased myocardium, ROS may be enhanced due to a defective scavenging system or an enhanced production, leading to oxidative stress (OS). Several studies [97,98,99,100,101] have revealed the mechanism of ROS-induced ROS-release (RIRR). They observed that local ROS injury may rapidly gather and get above a critical threshold to cause myocardial OS. During this process cardiac mitochondria react to an elevated ROS level by the production of more ROS, the so-called RIRR [98]. Enhanced mitochondrial-derived RIRR bursts influence RyR2 and SERCA2a activity, resulting in increased cytosolic Ca^2+^. Either regulated by the inner membrane anion channel (IMAC) or the Ca^2+^-dependent PTP, RIRR contributes to mitochondrial dysfunction with consequent generation of an arrhythmogenic substrate [102,103]. For extended information, Gambardella and colleagues summarize the role of mitochondria in the generation of cardiac arrhythmias [104].

In summary, mitochondrial dysfunction, including mitochondria-initiated cell death and altered mitochondrial Ca^2+^ handling, seem to be a hallmark of the failing heart [105]. Nevertheless, since the underlying mechanisms are not fully understood yet, the ability of mitochondria to compensate for extensive Ca^2+^ levels remains a matter of debate [94,106]. 

## 3. Novel Therapeutic Options for Calcium Mediated Ventricular Arrhythmias

As summarized in the previous sections and previous reviews [107,108,109,110], cellular Ca^2+^ imbalance is one of the main triggers for the generation of malignant ventricular arrhythmias [111]. Therefore, various approaches have focused on restoring physiological Ca^2+^handling in the diseased heart.

Approximately 50 years ago, Fleckenstein [112] was the first to review Ca^2+^ channel blockers as new drugs for the treatment of coronary diseases. Indeed, drugs affecting intracellular Ca^2+^handling have the ability to decrease automaticity of Ca^2+^ related AP generation and have emerging uses as anti-arrhythmic agents. Classical drugs that target proteins responsible for cellular Ca^2+^-handling include β-blockers (inhibitor of β-AR activation) [113], and the LTCC inhibitors verapamil [114] and diltiazem [115,116], as well as the class I anti-arrhythmic flecainide. Although this agent is an anti-arrhythmic drug with Na^+^ channel blocking properties, in mouse models and patients with CPVT it has been shown that it also inhibits RyR2 [117,118].

Unfortunately, current anti-arrhythmic drug therapy often increases or ideally has a neutral effect on cardiac-related mortality [5,119,120,121,122,123]. While device therapies, such as implantable cardioverter defibrillators (ICDs), have a positive effect on correct heart rhythm, they do not treat or cure the cause of arrhythmia. Of note, they change the electrical signaling to convert tachyarrhythmias following their onset. Besides its preventative effects, ICD implantation may lead to psychopathological disorders and may therefore reduce patient’s quality of life. Furthermore, several surgical complications are associated with ICD implantations, as well as device and lead failures [124]. 

Due to the reasons mentioned above and in order to achieve a potential curative effect, a high demand for novel therapies to treat the origin of ventricular arrhythmia is present. In this review, we want to summarize novel therapeutic strategies and targets to combat Ca^2+^ related VT, such as gene therapy. Of note, by using various viral vectors (Table 1) this experimental approach is able to target specifically arrhythmia related proteins. Therefore, cardiac gene therapy is one of the emerging therapeutic approaches that can offer a new therapeutic strategy. Indeed, various experimental studies have already investigated the potential effects of gene therapy aimed to target complexes involved in Ca^2+^ related ventricular arrhythmias.

### 3.1. Gene Therapy Targeting LTCC

Due to their correlative effect on contractility or blood pressure, in patients suffering from ventricular arrhythmias and HF, strategies of blocking LTCC are not optimal yet [142]. For this reason, gene therapy to shorten APD without impairing influences on systemic physiological behavior is intended. Table 2 lists up gene therapy strategies targeting LTCC in vivo and in vitro.

To the best of our knowledge, Muarata et al. [143] was the first to create a biological effective genetic Ca^2+^ channel blocker by the overexpression of the ras-related small G-protein Gem. This was achieved by adenovirus (AD) mediated gene transfer of Gem, resulting in a significant decreased LTCC current density in ventricular myocytes. Consequently, this approach successfully promoted shortening of APD with consequent abbreviation of electrocardiographic QTc interval [143]. In 2007, Cingolani et al. [144] seized on this topic and created a genetically knockdown of LTCC accessory β-subunit gene by a short hairpin RNA template sequence. In vivo injection of the lentiviral vector with the RNA template partially inhibited LTCC current and reduced Ca^2+^ transient amplitude in neonatal rats. Furthermore, the hypertrophic response in vivo and in vitro was attenuated, without affecting systolic performance [144].

As expression dependent manipulation of large cardiac genes (>6 kb) is difficult, due to the limited packaging capacity of viral vectors (4–7 kb), improvements of the delivery approaches were anticipated. In their elegant proof of concept study, Subramanyam et al. [145] developed a new technique targeting the 6.6 kb pore-forming α_1C_-subunit of LTCC. Concerning this, the so-called split-intein protein transsplicing was used. Split-intein-tagged α_1C_ fragments encoding dihydropyridine-resistant channels were incorporated into AD and applied to adult rat cardiomyocytes in vitro. Of note, triggered recombinant LTCC channel expression promoted Ca^2+^ transients and supported β-adrenergic regulation of excitation-contraction coupling [145].

### 3.2. SERCA2a Gene Therapy

In diseased heart, Ca^2+^ reuptake by SERCA2a is commonly decreased and causes major defects in excitation-contraction coupling [146]. Since SERCA2a is the major pump that transfers cytosolic Ca^2+^ back to the sarcoplasmic reticulum during diastole, promoting SERCA2a expression seems to be a promising goal in HF. Indeed, multiple preclinical studies using SERCA2a adeno-associated virus (AAV) mediated gene transfer [45,147,148,149] encouraged broader application of this approach in the HF population. However, while various preclinical trials [42,48,49,50,51] supported beneficial outcomes mostly by restoring mechanical function, the CUPID2 trial, most probably due to technical problems with gene delivery, failed to elucidate therapeutic benefits in the clinical setting [59]. Nevertheless, pathophysiology of intracellular Ca^2+^ handling is complex. Consequently, besides affecting the heart’s mechanical function, SERCA2a expression might also influence cardiac electrophysiology. Theoretically, the restoration of SERCA2a activity and prevention of cytotoxic Ca^2+^ overload could affect the generation of beat-to-beat repolarization alternans and consequent DADs. 

Nevertheless, it remains unclear whether an increase in SERCA2a expression by gene therapy could alter the electrophysiological substrate that triggers the generation of cardiac arrhythmias. Furthermore, SERCA2a overexpression dependent Ca^2+^ overload of the SR could promote a Ca^2+^ leak through RYR2 and contribute to an increased susceptibility to arrhythmias.

However, in contrast to these speculations, in rodent models, SERCA2a overexpression did not exacerbate SR Ca^2+^ leak through RYR2 and has suggested protection against, not promotion of, arrhythmias (Table 3) [150,151,152,153,154,155]. Indeed, Lyon et al. explored an established rat chronic HF model, which is characterized by a high burden of spontaneous ventricular arrhythmia [153]. In treated rats, SR Ca^2+^ load was stabilized, and arrhythmia was significantly reduced. Furthermore, Ca^2+^ leak decreased to a level not statistically different from control animals. Of note, RYR2 phosphorylation at Ser^2815^, which is accomplished by CaMKII, was lowered, resulting in an increase in the threshold for SR Ca^2+^ release with consequent lower rates of SCR events and cellular triggered activity in vitro and in vivo [153]. These findings are supported by previous data. Xie et al. reported a suppression of Ca^2+^ alternans in cultured rabbit CMs [156]. Other studies in healthy [152] but also in a pressure-overload induced HF model in the guinea pig [154] demonstrated prevention of Ca^2+^ and APD alternans with consequent suppression of pacing-induced ventricular arrhythmias. Furthermore, in small as well as large animal models the incidence of I/R arrhythmias was suppressed [150,151].

More recently, a novel mechanism how SERCA2a gene therapy might affect cardiac electrophysiology was explored [160]. In a large animal model, for the first time, the authors demonstrated that SERCA2a gene transfer could prevent cellular remodeling in an advanced stage of ischemic HF. The animals were treated by gene therapy after HF was confirmed, one month after myocardial infarction. Even in this pronounced stage, SERCA2a gene therapy prevented the incidence of dobutamine dependent ventricular arrhythmias in vivo. Importantly, while gene therapy was not related to improvements of hemodynamic function, SERCA2a related prevention of major electrophysiological remodeling processes was observed. SERCA2a expression increased conduction velocity reserve, likely by preventing CAMKII overactivation with consequent increase in the reserve of cardiac excitability. Of note, in HF electrophysiological remodeling promotes reduction of the conduction velocity reserve leading to an increased susceptibility for reentry mechanisms at higher pacing rates [162]. Consequently, in this study, HF dependent prolongation of QRS duration on ECG in vivo as well as the rate of pacing induced sustained VT and ventricular fibrillation (ex vivo) were decreased. Therefore, this study indicated primary effects of SERCA2a gene therapy on myocardial excitability, independently of altered mechanical function [160].

Strauss and colleagues [161] investigated potential effects of SERCA2a gene delivery on the arrhythmogenic substrate in pulmonary arterial hypertension (PAH). Of note, this potential, fatal pathology promotes right heart failure with consequent malignant electrophysiological remodeling. In their elegant study, male rats developed advanced PAH after subcutaneous injection of monocrotaline. This approach was followed by aerosolized delivery of AAV1/SERCA2a after three weeks. Indeed, delivery of SERCA2a ameliorated myocardial electrophysiological remodeling including fibrosis and altered ion channel expression. These findings coincided with increased rate of VT at rapid heart rates as well as major electrophysiological alterations in the non-treated PAH animals (including prolonged APD, increased APD heterogeneity, a reversal in the trans-epicardial APD gradient and marked conduction slowing), which was not described in the treated group. Thus, for the first time Strauss et al. depicted a non-cardiac gene delivery approach with successful suppression of ventricular arrhythmias [161]. 

### 3.3. Gene Therapy Targeting RyR2 Complex

CPVT is the most prominent form of genetic mutations concerning changes particularly in RyR2 and CASQ2. Mutations in RyR2 are mostly autosomal-dominant and lead to unzipping of CaM from RyR2 protein, thus repealing the inhibitory effect of CaM on RyR2. These results in SCRs, promoting arrhythmia and HF [19,163,164]. Indeed, a recent study demonstrated ryanodine receptor Ca^2+^ leak to act as a useful biomarker in patients with HF [165].

The RyR2 channel itself is targeted rarely by experimental gene therapy (Table 3), as several compounds have been developed to reduce Ca^2+^ leak through RyR2, such as JTV519 [166]. Nevertheless, two studies aimed RyR2 channel modification by gene therapy (Table 4), achieved by the new genome editing technique CRISPR/CAS9 (clustered regularly interspaced short palindromic repeats with caspase 9). As in CPVT RyR2 channels containing one mutant monomer can promote a Ca^2+^ leak, it is hypothesized that modification of even one allele, such as silencing or removing of a fraction of the mutant subunits, could restore cardiac physiology [167,168].

Consequently, Bongianino et al. investigated allele-specific gene silencing in the autosomal-dominant form of CPVT [167]. They used a short-interfering ribonucleic acid (siRNA) to specifically silence an allele of RyR2 that bears a dominant negative (R4496C) mutation in the CPVT mouse model (Table 4). Indeed, by using this elegant technique, the authors were able to reduce isoproterenol-induced DADs and triggered activity. Importantly, this was followed by a decreased susceptibility for adrenergically mediated VT. These findings were consistent with reverted ultrastructural changes of SR and transverse tubules, as well as attenuated mitochondrial abnormalities [167]. Similar findings were confirmed by others. CRISPR/Cas9 genomic allele silencing of R176Q was achieved by single subcutaneous injection at postnatal day 10. This highly specific editing normalized the incidence of Ca^2+^ sparks to normal levels and abolished the genesis of ventricular arrhythmias in the treated animals [168].

Successful results were also achieved by targeting regulatory proteins like CaMKII. Of note, CaMKII dependent phosphorylation is crucial for RyR2-mediated SR Ca^2+^ release. Consequently, pharmacological inhibition reduces Ca^2+^ handling abnormalities and arrhythmias in human iPSC-CMs [174]. Importantly, CaMKII phosphorylation of RyR2 at serine 2814 is evident to prevent the latent arrhythmic potential of RyR2 mutations in CPVT-1 indicating antiarrhythmic effects [73,175]. Therefore, efforts were started to explore the genetic inhibition of CaMKII in a CPVT-1 mouse model (RYR2^R176Q/+^) [173]. CaMKII inhibition was achieved by AAV9 gated CaMKII inhibitory peptide autocamitide-2-related inhibitory peptide (AIP), which was fused to a green fluorescent protein (GFP). The outcomes showed a robust expression of the peptide solely in the heart. This successful approach was able to prevent VTs either after adrenergic stimulation or programmed ventricular pacing [173]. These results were supported by concomitant in vitro studies in human induced pluripotent stem cells cardiomyocytes (iPSC-CMs), derived from two patients with different RyR2 mutations. In this cellular model CPVT-induced abnormal Ca^2+^ release events were efficiently suppressed by AIP [173]. 

Further studies have addressed the recessive form of CPVT, namely CPVT-2. This form is characterized by mutations in CASQ2, which is an essential regulating part of the RyR2 macromolecular complex. RyR2 channels lacking the CASQ2 protein tend to open spontaneously, without the need for LTCC dependent trigger Ca^2+^ influx [176]. Consequently, in animal models with mutant CASQ2 protein a higher frequency of Ca^2+^ dependent ventricular arrhythmias is observed [177]. In CPVT several missense, deletion or nonsense mutations, which result in a severe reduction, or complete loss of the protein have been detected [178]. Therefore, various studies have focused on restoring CASQ2 expression using diverse gene transfer techniques.

In iPSCs from a patient carrying the homozygous CASQ2-G112+5X mutation, AAV9 dependent gene delivery was able to normalize cellular Ca^2+^ transient and reduce the incidence of DADs [170]. These optimistic results are supported by further animal studies. Denegri and colleagues have investigated a murine CPVT knock-model of CASQ2 (CASQ2^R33Q/R33Q^ (R33Q) mutation) [169]. Successful transfection of the wild-type CASQ2 gene was achieved by an AAV9 vector (AAV9-CASQ2), which was delivered in neonatal mice on day 3. In vivo and in vitro investigations of this model have revealed astonishing curative effects of this therapeutic approach. Restoration of physiological expression and interaction of CASQ2, junctin and triadin was followed by normalization of electrophysiological and ultrastructural abnormalities of cellular Ca^2+^-handling. Importantly, due to successful transfection life-threatening arrhythmias have been abrogated [169]. Further findings in a murine CASQ2^D307H^ model supported these enthusiastic results. Interestingly, in the study by Kurtzwald-Josefson et al. [171] antiarrhythmic efficacy was dependent on the CASQ2 level expression (at least greater than 33% of normal CASQ2). 

Further studies have successfully focused on other regulatory proteins of the RyR2 complex, namely CaM. CaM gene transfer into the CPVT knock-in mouse model carrying a CASQ2^R33Q/R33Q^ (R33Q) mutation was also able to restore defective Ca^2+^ handling and prevent consequent isoproterenol-triggered VT [172].

### 3.4. Cardiac NCX Constituting a Target for Gene Therapy?

Although transgenic mouse models overexpressing NCX have been studied for a long time [179,180,181], NCX currently fails to be a direct and single target for efficient gene therapy approaches. This may be due to the great dependence of NCX function on several other channels, such as SERCA2a and LTCC. Overexpression of cardiac NCX results in cardiac hypertrophy and increases the risk for HF presumably by alterations in excitation-contraction coupling [182]. On the other hand, homozygous NCX-deficiency results in premature death of embryonic mice by days 9 and 10 due to defect pacemaker function [183]. Studies addressing targeted NCX gene transfer are summarized in Table 5.

As in most cardiac pathologies NCX activity seems to be reduced [189,190,191], restoration of the exchanger function might be an attractive therapy approach. However, therapeutic increase in NCX activity revealed adverse pathologically altered phenotypes. Due to potential dependence on Na^+^/K^+^-ATPase (NKA) activity, NCX overexpression resulted in systolic and diastolic dysfunction. These potential alterations observed in transgenic mouse model as well as in transfected CMs are probably related to decreased SR Ca^2+^ stores with concomitant cellular Ca^2+^ overload [184,187]. 

On the other hand, in some pathological manifestations decreased NCX activity is accompanied by reduction in SERCA2a function. Therefore, the usage of a dual therapy including simultaneous elevation of SERCA2a and NCX function seems favorable. Terracciano and colleagues [159] have investigated this approach by overexpressing cardiac NCX in a transgenic mouse model with decreased SERCA2a activity. They were able to conclude that overexpression of NCX may compensate SERCA2a inhibition and restore normal Ca^2+^ homeostasis [159]. 

In vitro studies downregulating NCX in rat cardiomyocytes were performed by Tadros et al. [186]. They showed a reduced cellular influx and efflux of Ca^2+^ that is contingent on reduced NCX function.

### 3.5. Mitochondrial Ca^2+^ Channels—Possible Targets for Gene Therapy

A new proof-of-principle study has been published by Gammage et al. [192]. This elegant trial addressed the question if mutations in the mitochondrial genome (mtDNA) could be corrected by mitochondrially targeted zinc finger-nucleases (mtZFNs). The authors successfully targeted specific mtDNA, which resulted in the restoration of molecular and physiological disease phenotypes in the heart tissue [192]. Consequently, mitochondria are becoming an attractive target for novel genetic therapy approaches.

As already described, mitochondria have the ability to transport huge amounts of cellular Ca^2+^. This vital process has major regulatory effects on mitochondrial function (generation of ATP, genesis of ROS, etc.) but also on the whole cellular metabolism including cellular Ca^2+^ handling [29,30,31,32]. Therefore, targeting proteins involved in mitochondrial Ca^2+^ handling seems to be a promising strategy to suppress cardiac arrhythmias. This concept is supported by several studies, in which MCU activity was suppressed via pharmacological or genetical approaches, resulting in a diminished myocardial damage after I/R [29,193]. Table 6 presents an overview of recent trials addressing genetic modifications in MCU and further mitochondrial proteins involved in mitochondrial Ca^2+^ handling. 

Wu et al. [194] were the first to create a transgenic MCU knock-out mouse. The group used an AD vector to transduce dominant negative (DN) MCU into mouse embryonic stem cells, which were further inserted into pseudo-pregnant females. DN-MCU mice showed normal resting heart rates. However, they were incapable of accelerating fight or flight heart rate acceleration. Thus, for the first time this elegant technique revealed chronotropic function of the MCU indicating this channel as a promising target for cardiac arrhythmias [194]. Others applied siRNA to target MCU in CMs. This approach prevented mitochondrial Ca^2+^ overload with consequent permeability pore opening, leading to reduced hypoxia-reoxygenation injury indicating a potential strategy for arrhythmia prevention [195]. 

Since mitochondrial Ca^2+^ uptake is decreased in diabetic cardiomyocytes [197,198], genetic therapies aimed to increase MCU expression in diabetic mice. For this purpose, AAV9-MCU was generated and injected into the jugular vein of diseased animals. Indeed, after 4–6 weeks, normalized MCU levels as well as a restoration of mitochondrial Ca^2+^ handling were detected. These findings support the idea that abnormal Ca^2+^ handling in CMs could be restored via MCU transgene expression [196].

Further studies explored the function of UCPs. As already mentioned, the isoforms UCP2 and UCP3 are suggested to influence MCU activity [45,46,47,48,49,50]. Consequently, in UCP2 knockout mice altered excitation contraction coupling with compensatory reduced LTCC-activity is observed [45,51]. Indeed, UCP2-dependent modulations promoted APD shortening, increased slope factor of action potential upstrokes and more hyperpolarized resting membrane potential leading to altered ECG parameters (PR and QRS as well as shortening of the QTc interval). Importantly, modifications of electrophysiology were followed by increased incidence of DAD and a higher susceptibility to Ca^2+^ mediated ventricular arrhythmias [51]. Therefore, this study identified mitochondrial proteins involved in Ca^2+^ handling as attractive targets to influence Ca^2+^ related ventricular arrhythmias.

Since HF is accompanied by alterations in Na^+^ and Ca^2+^ concentrations, the mitochondrial NCX (NCLX) may also constitute a promising target for gene therapy [34,55,56]. Indeed, chronic administration of the NCLX inhibitor CGP leads to cellular remodeling, fibrosis and alters the susceptibility for ventricular arrhythmias and SCD [199]. Consequently, preservation of physiological NCX expression and function seems evident. The elegant work of Luongo and colleagues [193] addressed this issue by deleting the gene encoding NCLX (*Slc8b1*). Due to mitochondrial Ca^2+^ overload, they observed the development of HF with consequent SCD. On the contrary, overexpression of NCLX resulted in increased mitochondrial Ca^2+^ clearance with reduced permeability transition formation and decreased tissue necrosis [193]. Therefore, enhancing NCLX function in diseased hearts might be a promising therapeutic strategy in the future.

## 4. New Targets for Ca^2+^ Mediated Ventricular Arrhythmias

Besides the genetic targets mentioned above, further regulatory proteins that mediate cardiac Ca^2+^ homeostasis could constitute a potential point of application for gene therapy. These potential therapeutic targets are summarized in the following paragraph.

The small ubiquitin-related modifier 1 (SUMO-1) was uncovered as a crucial regulatory part of the SERCA2a complex in HF. Under physiological conditions, SUMO-1 binds to SERCA2a in a process called SUMOylation affecting the ATP-binding domain of SERCA2a. SUMOylation is followed by an increased ATP-binding affinity of SERCA2a with concomitant stabilization of the protein. However, in HF, SUMO-1 expression is decreased. Therefore, multiple studies have aimed for the expression of SUMO using genetic strategies in various HF animal models [200,201,202]. Indeed, the authors were able to restore cardiac hemodynamic function, indicating overexpression of SUMO-1 as a promising strategy in HF [200,201,202]. Nevertheless, further trials need to investigate the impact of SUMO-1 on the incidence of cardiac arrhythmias.

The Ca^2+^ binding protein S100 calcium-binding protein A1 (S100A1) is a further important regulator of cardiac performance. It directly interacts with SERCA2a and RyR2. Thus, it contributes to improved Ca^2+^ handling and contractile performance. Consequently, it plays a crucial role in the pathophysiology of HF and might provide a novel therapeutic target for treating acute and chronic cardiac dysfunction [203]. As suspected, in various HF model genetic enhancing of S100A1 expression was able to improve contractile function. Importantly, this was achieved by normalization of physiological Ca^2+^ transients and SR load due to an increase in SR Ca^2+^ uptake and reduced SR Ca^2+^ leak [204,205,206,207] Therefore, this strategy might be suspected to provide a potential tool to prevent Ca^2+^ related ventricular arrhythmias.

The regulation of Ca^2+^ influx into the cytoplasm seems not only to be limited to activation of the LTCC. Interestingly, when Ca^2+^ is depleted from intracellular stores, such as ER, store-operated Ca^2+^ entry channels (SOC) open. The most important SOC channel is the Ca^2+^ release-activated Ca^2+^ (CRAC) channel, which consist of multimers of ORAI (Ca^2+^ release-activated Ca^2+^ channel protein) family proteins, there of the best characterized is ORAI1. ORAI1/CRAC channels are regulated by the stromal interaction molecule 1 (STIM1). Of note, STIM1 is an ER transmembrane protein, which is activated through ER Ca^2+^ store depletion [208,209,210,211,212,213]. Through its effects on cellular Ca^2+^ cycling this protein is involved in various cellular pathologies including cancerogenesis [214,215,216,217,218,219,220,221,222,223]. However, investigations have also revealed regulatory function in several important cardiac physiological and pathophysiological processes including angiogenesis, pacemaker function and cardiac hypertrophy [218,219,223,224,225,226,227]. Therefore, this protein might constitute an interesting target for cardiac gene therapy including prevention of Ca^2+^ mediated ventricular arrhythmias [218,224,228,229,230]. 

Further potential gene therapy targets could consist of the Ca^2+^ regulating proteins PLN [231,232,233,234], junctophilin- (JPH2) [235,236,237] and the inhibitor of protein phosphatase 1 (I-1c) [238]. In experimental models all have been shown to improve cardiac Ca^2+^ handling and cardiac performance.

## 5. Conclusions

A defect in Ca^2+^ handling is one the major reasons for congenital but also acquired malignant ventricular arrhythmia syndromes. In recent years, major progresses were achieved to understand the underlying mechanisms of Ca^2+^ mediated arrhythmogenesis. Consequently, various cellular and mitochondrial targets were identified. The strategy of gene transfer is able specifically to aim disease relevant proteins. Indeed, targeting cardiac Ca^2+^ handling by somatic gene transfer has been demonstrated to be effective in experimental models. Therefore, various studies have explored this strategy to prevent Ca^2+^ related arrhythmogenesis. By targeting regulatory proteins of Ca^2+^ handling, these investigations were already able to uncover promising results by suppressing Ca^2+^ related congenital but also acquired arrhythmia mechanisms. Therefore, gene transfer to suppress Ca^2+^ related ventricular arrhythmias seems to be an attractive strategy in in the future. Nevertheless, first investigations in humans will be necessary to uncover the efficiency of this therapeutic approach. 

## Figures and Tables

**Figure 1 ijms-20-05304-f001:**
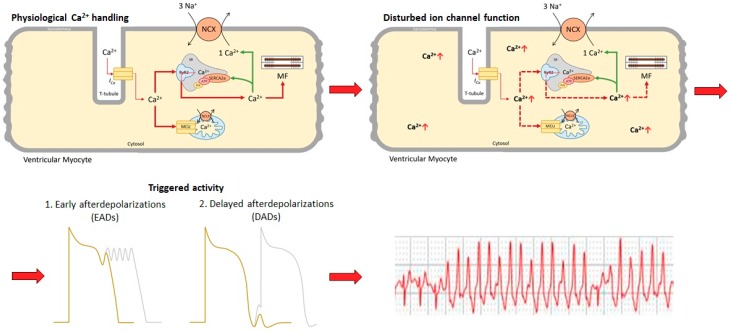
Calcium handling in cardiomyocytes illustrating the most important calcium-related organelles and channels. Disturbed ion channel functions or calcium mishandling results in an increased cytosolic calcium concentration. This further promotes the generation of early and delayed afterdepolarizations, leading to Ca^2+^-induced ventricular tachycardia. Ca^2+^: calcium; I_Ca_: calcium current; MCU: mitochondrial calcium uniporter; MF: myofilament; NCLX: mitochondrial potassium-calcium exchanger; NCX: sodium-calcium exchanger; RyR2: ryanodine receptor type 2; VES: ventricular extrasystoles; VT: ventricular tachycardia.

**Table 1 ijms-20-05304-t001:** Advantages and limitations of different gene delivery vectors and CRISPR/Cas9 system.

Method	Advantages	Limitations	References
Lentivirus	Broad host range Infection of dividing and non-dividing cells Low cytotoxicity Long-term expression (integration into genome) Insert capacity: 8 kb	No specific tropism for CMs (requires direct injection into the heart) Risk of insertional mutagenesis (integration into genome)	[125,126,127,128,129]
AD	Broad host range High level of gene expression No host genome integration Insert capacity: <35 kb	Short-term expression Strong immunogenicity No specific tropism for CMs	[127,129,130,131]
AAV	Relatively broad host range Low pathogenicity and toxicity Infection of dividing and non-dividing cells Long-term expression Serotype modulation for organ specificity	Difficulties in high transgene expression Delayed expression Presence of NAbs Small insert capacity of <5 kb	[127,129,132,133,134,135,136]
CRISPR/Cas9	Targeting specific DNA sequences Any organism Simple and precise (compared to gene targeting) Inactivation, integration and allele substitution possible Reactivation of non-dividing cells Low immunogenicity	Difficulties in off-target effects (nonspecific and mismatched genetic modifications) Difficulties in delivery of large Cas9 sequences	[137,138,139,140,141]

AD: adenovirus; AAV: adeno-associated virus; CRISPR/Cas9: clustered regularly interspaced short palindromic repeats with caspase 9; CMs: cardiomyocytes; DNA: deoxyribonucleic acid; kb: kilobases; NAbs: neutralizing antibodies.

**Table 2 ijms-20-05304-t002:** Gene therapies for ventricular tachycardia targeting L-type Ca^2+^ channel (LTCC).

Author (year)	Vector	Delivery Technique	Genetic Information	Species/Model	Outcomes
Murata et al. (2004) [143]	AD	Injection into LV cavity	Mutant Ras-related G-protein Gem W296G	Guinea pig/wt	↓ I_Ca-L_ in CMs ↓ QT in vivo
Cingolani et al. (2007) [144]	Lentivirus	Injection into LV cavity	Hairpin RNA for β_2_	Rat/aortic-banded model of LV hypertrophy	↓ I_Ca-L_ in CMs
Subramanyam et al. (2013) [145]	AD	In vitro	Split-intein-tagged α_1C_-fragments	Rat/wt	↑ Ca^2+^ transients β-adrenergic regulation

The origin of the genetic material used in the studies is indicated, if mentioned in the publication. Injection into the aortic root or the left ventricle (LV) cavity was performed during transient cross-clamping of the great vessels. Intramyocardial injection was performed after thoracotomy. ↓: decrease; ↑: support; AD: adenovirus; Ca^2+^: calcium; CMs: cardiomyocytes; I_Ca-L_: L-type calcium current; LV: left ventricle; RNA: ribonucleic acid; VM: ventricular myocard; wt: wild-type.

**Table 3 ijms-20-05304-t003:** Gene therapies to restore cardiac electrophysiology and to prevent ventricular tachyarrhythmias targeting sarco/endo plasmic reticulum calcium ATPase (SERCA2a).

Author (year)	Vector	Delivery Technique	Species/Model	Outcomes
Giordano et al. (1997) [157]	AD	In vitro	Rat— ↓ SERCA2a expression	↑ SERCA2a expression ↓ Ca^2+^ transients ↑ SR Ca^2+^ uptake
Hajjar et al. (1997) [158]	AD	In vitro	Rat— wt	↑ peak Ca^2+^ release ↓ resting Ca^2+^ levels
Terracciano et al. (2002) [159]	AD	In vitro	Rabbit— wt	↓ APD ↑ SR Ca^2+^ content
del Monte et al. (2004) [150]	AD	Intramyocardial injection	Rat— wt	↓ VT after I/R
Prunier et al. (2008) [151]	AD	Anterograde coronary injection	Swine— wt	↓ VT after I/R
Cutler et al. (2009) [152]	AD	Injection into aortic root	Guinea pig— wt	↓ APD alternans in vitro and ex vivo ↓ VT ex vivo
Lyon et al. (2011) [153]	AD AAV9	Intramyocardial (AV), or tail vein (AAV9)	Rat— HF	↓ VT ex vivo ↓ spontaneous and isoproterenol triggered VT in vivo
Cutler et al. (2012) [154]	AAV9	Injection into aortic root	Guinea pig— HF	↓ APD alternans ↓ VT ex vivo
Motloch et al. (2018) [160]	AAV1	Intracoronary injection	Swine— MI	↓ QRS duration in vivo ↓ VT in vivo and ex vivo
Strauss et al. (2019) [161]	AAV1	Aerosolized	Rat— PAH	↓ VT in vivo ↓ APD duration Reversed spatial APD heterogeneity ↑ Electrophysiological remodeling

The origin of the genetic material used in the studies is indicated, if indicated in the publication. Injection into the aortic root or the LV cavity was performed during transient cross-clamping of the great vessels. Intramyocardial injection was performed after thoracotomy. ↓: decrease; ↑: improvement; AD: adenovirus; AAV1: adenovirus-associated virus serotype 1; AAV9: AAV serotype 9; APD: action potential duration; EF: ejection fraction; HF: heart failure; I/R: ischemia reperfusion; LV: left ventricle; MI: myocardial infarction; PAH: pulmonary arterial hypertension; SERCA2a: sarcoplasmic reticulum Ca^2+^ ATPase 2a; UV: upstroke velocity; VT: ventricular tachycardia; wt: wild-type.

**Table 4 ijms-20-05304-t004:** Gene therapies for ventricular tachycardia targeting ryanodine receptor 2 (RyR2) and genes of RyR2-mediating proteins.

Author (year)	Vector	Delivery Technique	Genetic Information	Species/Model	Outcomes
Bongianino et al. (2017) [167]	AAV9	Intraperitoneal injection	miRyR2-U10	Mouse/wt	↓ DADs ↓ VT in vivo
Pan et al. (2018) [168]	AAV9	Subcutaneous injection	RyR2	Mouse/CPVT (R176Q/+)	↓ arrhythmias in vivo
Denegri et al. (2014) [169]	AAV9	Intraperitoneal injection	CASQ2	Mouse/CPVT (R33Q)	↓ VT in vivo
Lodola et al. (2016) [170]	AAV9	In vitro	CASQ2	Human/CPVT; iPSCs (CASQ2-G112+5X)	↓ DADs ↑ Ca^2+^ transient amplitude and duration of Ca^2+^ sparks
Kurtzwald-Josefson et al. (2017) [171]	AAV9	Intraperitoneal injection	CASQ2	Mouse/CPVT (CASQ2^D307H^ or CASQ2^Δ/Δ^)	↓ VT in vivo
Liu et al. (2018) [172]	AAV9	Intra-thoracic cavity injection	CaM	Mouse/CPVT (R33Q)	↑ Ca^2+^ handling ↓ VT in vivo
Bezzerides et al. (2019) [173]	AAV9	Subcutaneous injection In vitro	CaMKII	Mouse/ CPVT (RYR2^R176Q/+^) Human/CPVT; iPSCs (different mutations)	↓ ventricular arrhythmia in vivo

The origin of the genetic material used in the studies is indicated, if indicated in the publication. Injection into the aortic root or the LV cavity was performed during transient cross-clamping of the great vessels. Intramyocardial injection was performed after thoracotomy. ↓: decrease; ↑: improvement; AAV9: adeno-associated virus serotype 9; Ca^2+^: calcium; CaM: calmodulin; CaMKII: calmodulin-dependent protein kinase II; CASQ2: calsequestrin 2; CPVT: catecholaminergic polymorphic ventricular tachycardia; DADs: delayed afterdepolarizations; hiPSCs: human induced pluripotent stem cells; RyR2: ryanodine receptor type 2; VT: ventricular tachycardia; wt: wild-type.

**Table 5 ijms-20-05304-t005:** Downregulation or overexpression of cardiac Na^+^/Ca^2+^ exchanger (NCX).

Author (year)	Vector	Delivery Technique	Species/Model	Expression Properties	Outcomes
Schillinger et al. (2000) [184]	AD	In vitro	Rabbit/wt	OE	↓ contractile function
Terracciano et al. (2001) [175]	Transfection reagent	In vitro Injected into nuclei	Mouse/wt	OE	↑ Ca^2+^ handling and homeostasis
Ranu et al. (2002) [185]	AD	In vitro	Rabbit/wt	OE	↓ contraction amplitude
Tadros et al. (2002) [186]	AD	In vitro	Rat/MI	DR	↓ Ca^2+^ influx and efflux
Schillinger et al. (2003) [187]	AD	In vitro	Rabbit/wt	OE	Systolic and diastolic dysfunction
Bölck et al. (2004) [188]	AD	In vitro	Rat/wt	OE	↓ cell shortening at higher stimulation frequencies ↑ intracellular systolic Ca^2+^ and contractile amplitude at low stimulation rates

The origin of the genetic material used in the studies is indicated, if indicated in the publication. Injection into the aortic root or the LV cavity was performed during transient cross-clamping of the great vessels. Intramyocardial injection was performed after thoracotomy. ↓: decrease; ↑: improvement; AD: adenovirus; Ca^2+^: calcium; OE: overexpression; DR: downregulation; wt: wild-type.

**Table 6 ijms-20-05304-t006:** Gene therapies for HF and inherited cardiac diseases targeting mitochondrial proteins, such as mitochondrial calcium uniporter (MCU), mitochondrial uncoupling protein 2 (UCP2) and mitochondrial N^+^/Ca^2+^ exchanger (NCLX).

Author (year)	Vector	Delivery Technique	Genetic Information	Species/Model	Outcomes
Wu et al. (2015) [194]	AD	In vitro, Mouse embryonic stem cells	DN-MCU	Mouse/wt	MCU is necessary for physiological heart rate acceleration
Oropeza-Almazán et al. (2017) [195]	Transfection reagent	In vitro	siRNA targeting MCU	Rat/H/R injury	↓ mitochondrial permeability pore opening ↓ oxidative stress
Suarez et al. (2018) [196]	AAV9	Direct jugular vein injection	MCU	Mouse/Diabetic	Restoration of cardiac myocyte and heart function
Larbig et al. (2017) [51]	Knock-out model		UCP2^-/-^	Mouse/Knock-out	↓ I_Ca-L_ in CM ↑ slope factor of action potential upstrokes ↑ hyperpolarized resting membrane potential ↓ PR, WRS and QTc interval ↑ after-depolarizations ↑ arrhythmias
Luongo et al. (2017) [193]	Knock-out and OE model		*SLC8B1* (NCLX)	Mouse/Knock-out and OE	↑ *m*Ca^2+^ clearance Prevention of heart failure

The origin of the genetic material used in the studies is indicated, if indicated in the publication. Injection into the aortic root or the LV cavity was performed during transient cross-clamping of the great vessels. Intramyocardial injection was performed after thoracotomy. ↓: decrease; ↑: improvement; AD: adenovirus; AAV9: adeno-associated virus serotype 9; ATP: adenosine triphosphate; Ca^2+^: calcium; CM: cardiomyocyte; DN: dominant-negative; H/R: hypoxia/reoxygenation; MCU: mitochondrial calcium uniporter; OE: overexpression; siRNA: small interfering ribonucleic acid; VT: ventricular tachycardia; wt: wild-type.

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
