# Peer review of "Pathophysiology of Calcium Mediated Ventricular Arrhythmias and Novel Therapeutic Options with Focus on Gene Therapy"

_ijms, 2019, doi:10.3390/ijms20215304_

Round 1

Reviewer 1 Report

Paar et collaborators collect, exhaustively, the literature focused on pathophysiology of calcium mediated ventricular arrhythmias and the different therapeutic strategy based mainly on gene therapy. The first paragraph of the manuscript is dedicated to the role of calcium in ventricular cardiomyocytes (action potential, cardiac contraction) and the description of the different intracellular subunits involved in intracellular calcium handling. Afterwards, the authors focus their attention on the pathophysiology of calcium mediated arrhythmia describing causes and main actors. Last part of the paper provides a comprehensive and well commented list of papers proposing therapeutic options to fight/rescue calcium mediated ventricular arrhythmias. The manuscript is well written and easy to read. Tables are explicative and clear.

Just some minor comments/suggestions:

Page 2 line 53. I would add, for sake of clarity, the word ‘ventricular’ before ‘AP’: ‘Each ventricular AP starts …’ . Actually sinoatrial node AP are not generated by exactly the same set of ion channels than ventricular AP

Page 3, Line 106 and 108. The word ‘channel’ refers to ‘MCU’, am i right?  I would use MCU instead of channel.

Page3, from line 106 to 110. It would be possible to better explain or/and to add a citation on how UCP2 modulates cardiac excitation-contraction coupling via altered LTCC-activity?

Page3, line 159. ‘…exhibit a slight augmentation in response….’. To which augmentation do you refer to?

Author Response

Manuscript ID: ijms-605577

Type of manuscript: Review

Title: Pathophysiology of calcium mediated ventricular arrhythmias and novel therapeutic options with focus on gene therapy

Dear Reviewer 1:

Thank you very much for the highly favorable assessment of our manuscript. We are grateful to for the valuable input and enthusiasm for our work. In this revision, we addressed all the concerns and incorporated the major suggestions.

Sincerely yours,

Lukas J. Motloch

Revision of the reviewers´ comments:

Paar et collaborators collect, exhaustively, the literature focused on pathophysiology of calcium mediated ventricular arrhythmias and the different therapeutic strategy based mainly on gene therapy. The first paragraph of the manuscript is dedicated to the role of calcium in ventricular cardiomyocytes (action potential, cardiac contraction) and the description of the different intracellular subunits involved in intracellular calcium handling. Afterwards, the authors focus their attention on the pathophysiology of calcium mediated arrhythmia describing causes and main actors. Last part of the paper provides a comprehensive and well commented list of papers proposing therapeutic options to fight/rescue calcium mediated ventricular arrhythmias. The manuscript is well written and easy to read. Tables are explicative and clear.

We thank the reviewer for the favorable assessment of our work!

Just some minor comments/suggestions:

Page 2 line 53. I would add, for sake of clarity, the word ‘ventricular’ before ‘AP’: ‘Each ventricularAP starts …’ . Actually sinoatrial node AP are not generated by exactly the same set of ion channels than ventricular AP

Done!

Page 3, Line 106 and 108. The word ‘channel’ refers to ‘MCU’, am i right? I would use MCU instead of channel.

Thank you for this suggestion. The manuscript was edited.

Page3, from line 106 to 110. It would be possible to better explain or/and to add a citation on how UCP2 modulates cardiac excitation-contraction coupling via altered LTCC-activity?

This issue was addressed in the revised version of the manuscript.

Page3, line 159. ‘…exhibit a slight augmentation in response….’. To which augmentation do you refer to?

This issue was clarified.

Reviewer 2 Report

Although this reviewer warmly welcomes this manuscript, some issues should be addressed:

Please define "calcium leak".

The Authors should mention the latest updates in terms of RyR structure and function discussed in this recent paper from Joachim Frank's Laboratory (Subcell Biochem 2018), who was among the first groups to solve RyR structure.

Some References should be updated (e.g. #16 dated 2002 should be updated with the following recent report discussing the same topic: doi: 10.1007/5584_2017_106).

Calcium leak through RyR has been recently demonstrated to be useful as a biomarker in patients with heart failure.

The functional role of mitochondrial calcium in arrhythmogenesis (doi: 10.1007/978-3-319-55330-6_10) should be discussed.

A recent paper summarizing the latest advances in the research on intracellular calcium release channels (doi: 10.1113/JP272781) should be mentioned.

It is advisable to the Authors to incorporate a pictorial or cartoon representation of the topic of the review in order to facilitate the comprehension and increase the overall impact of the manuscript.

Author Response

Manuscript ID: ijms-605577

Type of manuscript: Review

Title: Pathophysiology of calcium mediated ventricular arrhythmias and novel therapeutic options with focus on gene therapy

Dear Reviewer 2:

Thank you very much for the highly favorable assessment of our manuscript. We are grateful for the valuable input and enthusiasm for our work. In this revision, we addressed all the concerns and incorporated the major suggestions.

Sincerely yours,

Lukas J. Motloch

Revision of the reviewers´ comments:

Although this reviewer warmly welcomes this manuscript, some issues should be addressed:

We thank the reviewer for his/her kind remarks and enthusiasm for our work.

Please define "calcium leak".

“Calcium leak” was defined.

The Authors should mention the latest updates in terms of RyR structure and function discussed in this recent paper from Joachim Frank's Laboratory (Subcell Biochem 2018), who was among the first groups to solve RyR structure.

We thank the reviewer for this productive suggestion. Indeed, the work by Santulli and colleagues is a very elegant summary on the functional elements, gating and activation mechanisms of the RyR, which was included in our work.

Some References should be updated (e.g. #16 dated 2002 should be updated with the following recent report discussing the same topic: doi: 10.1007/5584_2017_106).

We thank the reviewer for this important advice. The reference list was updated.

Calcium leak through RyR has been recently demonstrated to be useful as a biomarker in patients with heart failure.

Indeed, in their elegant work Kushnir and colleagues studied SR calcium leak in B-lymphocytes from heart failure patients as well as a heart failure mouse model. They were able to uncover a chronic intracellular calcium leak. Consequently, they presented a novel biomarker for monitoring the response to pharmacological and mechanical therapy in chronic heart failure patients. This important work was included in the revised version of our manuscript.

The functional role of mitochondrial calcium in arrhythmogenesis (doi: 10.1007/978-3-319-55330-6_10) should be discussed.

This important work was added to our manuscript.

A recent paper summarizing the latest advances in the research on intracellular calcium release channels (doi: 10.1113/JP272781) should be mentioned.

We added this interesting publication to our work.

It is advisable to the Authors to incorporate a pictorial or cartoon representation of the topic of the review in order to facilitate the comprehension and increase the overall impact of the manuscript.

We thank the reviewer for this productive suggestion. A figure with a representative pictorial representation of the topic was added.

Round 2

Reviewer 2 Report

_